# Data Interpretation in Structural Health Monitoring: Toward a Universal Language

**DOI:** 10.3390/s25103054

**Published:** 2025-05-12

**Authors:** Magda Ruiz, Óscar Gualdrón, José A. Peral Mondaza, Luis Eduardo Mujica Delgado

**Affiliations:** 1Departament de Matemàtiques, Escola d’Enginyeria de Barcelona Est (EEBE), Universitat Politècnica de Catalunya (UPC), Campus Diagonal-Besòs (CDB), Carrer Eduard Maristany, 6-12, San Adrià de Besòs, 08930 Barcelona, Spain; luis.eduardo.mujica@upc.edu; 2Go Advice and Consulting, Calle 99 11b-66, Bogotá 110221, Colombia; oscar.gualdron@go-ac.co; 3Departament de Promoció Econòmica, Ajuntament de Sant Andreu de la Barca, Escoles Velles, Carrer Ctra. de Barcelona, 1, Sant Andreu de la Barca, 08740 Barcelona, Spain

**Keywords:** Structural Health Monitoring (SHM), monitoring, diagnosis, diagnostic reliability, AI and machine learning in SHM, data interpretation, data standardization, subjectivity in data, linguistic framework

## Abstract

Structural Health Monitoring (SHM) relies on the effective communication between sensors and diagnostic systems, yet data interpretation remains inconsistent and subjective. This paper introduces a novel perspective, viewing data as a form of language with its own syntax, semantics, and pragmatics. By adopting this linguistic framework, the study emphasizes the need for standardized “grammars” in data collection, processing, and analysis to reduce ambiguity and enhance diagnostic reliability. Using case studies from SHM, the paper illustrates how subjective decisions in variable selection, cluster labels, preprocessing, and modeling introduce biases that affect the outcomes. The findings highlight the potential of context-aware algorithms and integrated data sources to mitigate these biases. This conceptual approach has broader implications for data science, suggesting a universal “language of data” that fosters consistency and collaboration across disciplines. By recognizing the constructed nature of data, this work offers a path toward more accurate, efficient, and reliable structural diagnostics, advancing both SHM practices and data interpretation methodologies.

## 1. Introduction: Toward a Standardized Data Language in SHM and Beyond

Structural Health Monitoring (SHM) is an essential field dedicated to ensuring the safety, performance, and longevity of critical infrastructure. SHM involves the continuous assessment of structures using sensors and diagnostic systems. However, a persistent challenge in this domain lies in the effective communication of information between these systems. Data, while often regarded as purely technical, carries inherent ambiguities introduced by human decisions, from variable selection and cluster labeling to interpretation methods. These ambiguities are not unique to SHM; they extend across all fields of data analysis and artificial intelligence, where inconsistencies in labeling, variable naming, and data structuring hinder the interoperability of models and the reliability of diagnostics.

One significant source of ambiguity is the arbitrariness in labeling data clusters. Categories assigned to similar structural conditions are often assumed to be equivalent across systems or practitioners. However, these classifications are shaped by expert interpretation, contextual criteria, and domain-specific conventions, leading to variations in diagnostic labeling, while such variability does not always result in outright inconsistencies, it introduces an element of subjectivity that can impact comparability and interoperability between different SHM frameworks. This lack of standardization complicates the development of generalizable models and hinders the integration of AI-driven diagnostic tools.

This challenge mirrors broader issues in data science and AI, where the absence of a unified framework for data structuring and interpretation results in inconsistencies across disciplines. Addressing this issue requires a deeper understanding of how categories are constructed and standardized within SHM and data-driven decision-making systems.

Inspired by parallels with natural language, this paper proposes a conceptual framework that views SHM as a *communicative process*, where data must adhere to a shared “grammar” to ensure the clear and accurate communication between sensors, models, and decision-making systems. Just as effective language relies on syntax, semantics, and pragmatics, structured data analysis requires a well-defined framework to standardize class labeling, variable naming, sensor system homogenization (data collection for database creation), and communication between sensors and AI models.

To better illustrate the scope of these challenges and the potential benefits of a linguistic approach to data structuring, Table 1 summarizes key issues affecting data science and how a structured data language could address them.

As shown in Table 1, the lack of a structured data language affects various aspects of data science, from bias in machine learning to the replicability of research findings. The introduction of a linguistics-based framework, which incorporates syntax, semantics, and pragmatics into data representation, provides a systematic approach to standardizing information across disciplines. By reducing inconsistencies in labeling, variable naming, and data structuring, this framework enhances interoperability between systems, improves model generalization, and fosters more reliable AI-driven decision-making.

Thus, the objectives of this work are threefold as follows:To address the critical challenge of effective data communication in SHM, framing it as a problem of linguistic coherence.To propose a perspective on data analysis, treating it as a language with key linguistic elements—syntax, semantics, and pragmatics.To outline a standardized protocol for data interpretation, reducing ambiguities, including those from inconsistent labeling, and improving the reliability of structural diagnostics.

This work extends beyond traditional SHM practices by integrating insights from linguistics, constructivism, and data science. By recognizing and addressing the role of arbitrariness in data categorization, it aims to establish a **universal “language of data”**, not only for SHM but for all domains where structured data interpretation plays a critical role. This shift toward a linguistically inspired data framework has the potential to enhance interoperability, improve model generalization, and pave the way for more accurate, efficient, and reliable AI-driven assessments across disciplines.

## 2. Contributions and Novelty

The study of Structural Health Monitoring (SHM) has significantly evolved, with various methodologies focusing on data-driven diagnostics, feature selection, and signal processing. However, a persistent challenge remains in the standardization of data communication, particularly in how structural conditions are labeled, categorized, and interpreted across different systems. Previous studies have examined aspects of SHM standardization, but most efforts have been fragmented, lacking a unified linguistic framework that ensures semantic consistency across methodologies and disciplines.

This work introduces a novel perspective by treating SHM data as a structured language, applying linguistic principles to resolve inconsistencies in data interpretation. Unlike traditional approaches that focus solely on statistical pattern recognition or sensor fusion, this study proposes a universal grammar that governs data structuring, interpretation, and communication between AI models and human experts.

The key contributions of this work are outlined as follows:A linguistic framework for SHM: Drawing parallels with natural language, this work formalizes SHM data processing in terms of syntax, semantics, and pragmatics, providing a structured approach to reduce diagnostic ambiguity.A standardized lexicon for structural conditions: A universal classification protocol is proposed to improve the interoperability of AI models across different monitoring systems.A conceptual approach to category standardization: The study highlights the impact of subjective labeling and proposes methods to align classification criteria across disciplines, ensuring consistency in diagnostics.

By integrating these contributions, this work advances the replicability and automation of AI-based SHM while addressing a long-standing gap in data standardization. The adoption of this approach could facilitate interdisciplinary communication, improve the reliability of damage detection, and enhance AI-driven decision making in real-world SHM application.

## 3. A New Perspective: Data as Language in SHM and Beyond

In the rapidly evolving fields of Structural Health Monitoring (SHM) and data science, data are the cornerstone of analysis and decision-making. However, how we approach and interpret data remains inconsistent, often shaped by the subjective decisions of researchers and practitioners. This lack of standardization presents a critical challenge outlined as follows: while natural language has rules and structures that allow for universal understanding, the “language of data” remains fragmented and specific to the researcher.

This paper introduces a perspective that seeks address this challenge by treating data as a form of language. Just as natural language relies on shared grammar, syntax, and context for effective communication, data in SHM and data science can be structured and interpreted using linguistic principles. This approach is not just a theoretical exercise; it aims to create a standardized framework for understanding and interpreting data, minimizing ambiguity and misinterpretation.

One of the significant challenges in achieving such standardization is the arbitrariness inherent in data labeling, particularly for clusters representing similar structural conditions. While variables and sensors contribute to the framework of data collection, the categories assigned to clusters play an equally crucial role in shaping diagnostics. Arbitrary or non-equivalent labeling can lead to divergent interpretations of similar data, reducing the reliability and comparability of diagnostics. Addressing this issue requires integrating category standardization into the proposed linguistic framework to ensure consistent interpretations across systems.

In SHM, where the stakes are high (economic losses, environmental damage, and mainly loss of human lives), the need for this framework is particularly pressing. From detecting damage in infrastructure to localizing faults and predicting their progression, the effectiveness of SHM systems relies heavily on how data are collected, processed, labeled, and analyzed. Yet, the decisions regarding which variables to monitor, how to preprocess the data, and what methods to use for interpretation—along with how clusters are labeled—are often subjective. These choices introduce biases that can affect the accuracy and reliability of diagnostics.

By conceptualizing data as a language, this work opens the door to a more structured and standardized approach. The objective is clear—to enable researchers and practitioners to “speak the same language” when it comes to data, improving consistency and communication across disciplines. While the primary focus of this paper is SHM, the implications extend far beyond, offering valuable insights for data science as a whole.

This perspective invites readers to consider data not just as numbers or measurements but as a form of communication that requires a shared understanding to be truly effective. It challenges traditional notions of data as purely objective and instead highlights its inherently constructed nature. By adopting this linguistic lens, the inconsistencies and biases that hinder progress in SHM and data science, including those introduced by arbitrary labeling, can begin to be addressed, paving the way for more reliable solutions.

The following sections will delve into the methodology and case studies that illustrate this concept in action, demonstrating how the “language of data” can transform both theory and practice.

## 4. Parallelism Between Natural Language and Data Language

In Structural Health Monitoring (SHM), drawing comparisons to natural language may provide a useful framework for understanding how information is collected, processed, and interpreted. Considering data analysis as a form of “language” suggests that data might not be purely neutral or objective entities but rather a structured medium that can be analyzed and understood in ways similar to human communication. This perspective invites further exploration into the potential parallels between these fields and their implications for SHM practices. In the same way, the concept of data functioning as a “language” in SHM mirrors the components of natural language, such as syntax, semantics, pragmatics, and arbitrariness. This analysis is supported by the works of Mujica and Ruiz [1,2,3], as well as their students, who applied these ideas to practical SHM tasks.

### 4.1. Syntax: Rules That Shape Monitoring

In both natural language and Structural Health Monitoring (SHM), syntax provides the framework for structuring information. According to linguistic theories like those of Rafael Echeverría in Language ontology, syntax is the foundation for creating coherent and understandable communication [4]. Similarly, in SHM, syntax governs how data are collected, organized, and processed to convey meaningful insights about the health of a structure [1,5,6].

### 4.2. Grammatical Structures and SHM Syntax

In natural language, as noted by Echeverría, grammar provides the rules that ensure coherence and structure in communication, encompassing declarative, interrogative, and imperative forms [4]. In SHM, these forms translate into how data are collected and interpreted, as follows:-Declarative syntax (data collection protocols): In SHM, data collection follows a structured set of protocols, much like declarative sentences in language. For example, the placement of sensors in a UAV’s fuselage at stress-critical points ensures that the system can “speak” clearly about its health. The data are gathered in a pre-determined manner, similar to how declarative syntax structures factual statements [1].-Interrogative syntax (diagnostic testing): When testing a structure, engineers often simulate specific scenarios, much like asking a question in natural language. The system responds with data, and the engineers must “decode” the response to assess the structure’s condition. This process mirrors interrogative syntax in language, where a question seeks information [1,6].-Imperative syntax (real-time monitoring): SHM systems often require immediate feedback, similar to imperative commands in language. Sensors must continuously report on the structure’s health, providing real-time data to ensure immediate detection of structural anomalies [1,7].

### 4.3. Vocabulary, Lexicon, and Neologisms

In the same way that a lexicon provides a set of words for constructing sentences, the selection of variables in SHM (e.g., stress, strain, and temperature) forms the vocabulary of data collection. According to Paul Watzlawick’s theory of Constructivism, the interpretation of reality is shaped by the language we use [8]. In SHM, the variables chosen determine the framework within which the data are interpreted.

-Building SHM vocabulary: Just as vocabulary in natural language evolves to accommodate new concepts, SHM expands its “vocabulary” with new variables and sensor technologies. The richer the vocabulary of data (more variables and sensors), the more comprehensive the understanding of the structure’s health [1,5].-Neologisms in SHM: Just as language evolves by introducing new words (neologisms), SHM introduces new variables or sensor types to capture emerging phenomena. For example, the introduction of micro-deformation sensors can be seen as adding new “words” to the SHM lexicon, allowing for more nuanced monitoring and analysis [1,8]. However, while variables and sensors are essential elements, the most significant source of interpretive divergence often lies in the categorization of data clusters. The arbitrary labeling of clusters can lead to non-equivalent diagnostic potentials, especially when similar structural conditions are classified differently. Standardizing these categories, similar to defining precise terms in a lexicon, is vital for reducing inconsistencies in diagnostics.

### 4.4. Cultural Dependency and Bias in Syntax

According to the *Joule linguistic model*, which emphasizes the influence of context on language interpretation, syntax and meaning are often culturally dependent [4]. Similarly, in SHM, the selection of variables to monitor and the interpretation of data is influenced by the background, training, and biases of the engineers. This parallels the cultural biases that shape the use and interpretation of language.

-Cultural influence in SHM: Different fields within engineering may prioritize different aspects of structural monitoring, just as different cultures emphasize different linguistic structures. For example, while aerospace engineers might focus on stress and fatigue in UAV structures, civil engineers monitoring bridges may prioritize load-bearing capacity and material deformation [7,9].

Echeverría’s language ontology proposes that language serves not only as a tool for describing reality but also for constructing it [4]. A similar notion applies to SHM, where the collected data and the analytical methods employed actively influence the understanding of a structure. The decisions made in designing the monitoring system effectively define the “reality” being communicated, introducing inherent biases within both language and SHM [4,8].

## 5. Methodology

Several methodologies can strengthen this work by illustrating the necessity for standardization and the adoption of a linguistic framework for interpreting data in SHM. These methodologies emphasize repeatability, objectivity, and clear communication. The following are some key methodologies and the way in which they support the main goal:

### 5.1. Principal Component Analysis (PCA)

Relevance: PCA reduces the dimensionality of datasets, retaining only the most critical variables. It mirrors the semantic process in language, where core meanings are distilled from complex sentences.Support: By showcasing PCA’s ability to highlight essential features, the paper demonstrates how a “grammar” of data can be created, reducing ambiguity in SHM.Example: Mujica et al.’s work on PCA-based damage indicators illustrates the role of dimensionality reduction in isolating significant patterns for consistent interpretation [10].

### 5.2. Statistical Process Control (SPC)

Relevance: SPC techniques, such as control charts, are essential for identifying outliers and understanding process variability. These methods introduce standardized protocols for monitoring changes in structural health.Support: The use of SPC emphasizes the need for universal criteria to distinguish normal variations from critical anomalies, mirroring linguistic syntax.Example: Ruiz and Mujica’s application of SPC in monitoring multi-sensor systems highlights the value of consistent thresholds in diagnostics [3].

### 5.3. Machine Learning Techniques

Relevance: Algorithms like K-Nearest Neighbors (KNNs) and Support Vector Machines (SVMs) classify structural states by identifying patterns in the data. They demonstrate how “meaning” can be assigned to numerical inputs.Support: Machine learning exemplifies the integration of semantics into SHM by providing standardized interpretations of complex datasets, showing the need for a shared “vocabulary” in data analysis.Example: Ruiz’s use of supervised learning models in aeronautical SHM illustrates how classification algorithms can standardize diagnostics [1].

### 5.4. Context-Aware Algorithms

Relevance: Pragmatics in language depend on context; similarly, SHM requires models that incorporate environmental conditions, such as load or temperature, into their analyses.Support: Highlighting context-aware algorithms strengthens the argument for a shared “pragmatics” in SHM, ensuring diagnostics account for external variables.Example: Mujica’s studies on integrating environmental factors into PCA-based diagnostics illustrate the importance of context in reliable assessments [10].

### 5.5. Case-Based Reasoning (CBR)

Relevance: CBR uses historical cases in SHM to interpret new structural health data, similar to how prior experiences guide decision-making in language interpretation.Support: The inclusion of CBR emphasizes the potential for developing “contextual databases” in SHM that enable shared learning across different structures and monitoring systems.Example: Mujica et al.’s application of CBR in structural health monitoring demonstrates how leveraging historical data can enhance fault detection and diagnostic accuracy [11].

### 5.6. Multi-Sensor Data Fusion

Relevance: Combining data from multiple sensors ensures a comprehensive view of structural health, similar to integrating multiple linguistic cues for better communication.Support: Multi-sensor fusion illustrates the necessity for a shared syntax and semantics to interpret diverse data streams consistently.Example: Ruiz’s work on guided wave analysis for UAV structures demonstrates the importance of coordinated sensor networks in SHM [1].

## 6. Case Studies

To illustrate the application of the proposed framework, this section explores student projects supervised by Mujica and Ruiz, analyzing their methodologies within the context of the “language of data”.

### 6.1. SHM for UAV Fuselage

Syntax—The Grammar of Data Collection: This work prioritized stress and defect location as the key variables, mirroring the syntactic structure of a sentence. This structured approach defined the data collection process and subsequent interpretations [12].

Extracting Meaning from Data: PCA reduced the dataset’s dimensionality, isolating critical features. However, this exclusion of certain variables introduced semantic biases, underscoring the need for standardized protocols [1].

### 6.2. Torsional Wave Detection

Syntax—Sensor Placement: Sensor arrangement mirrored syntactic structuring, influencing how data were collected and interpreted [6].

Pragmatics—Subjectivity in Preprocessing: Filtering and bootstrapping steps reflected pragmatic decisions, emphasizing the impact of subjective preprocessing choices on diagnostics [3].

### 6.3. Aeronautical Structure Monitoring

Semantics—Assigning Meaning to Data: The use of KNN and PCA demonstrated the semantic process of categorizing data, similar to assigning meaning in language [7].

Neologisms—Creating New Tools: Adapting algorithms for specific aeronautical conditions reflected the creation of “neologisms”, expanding the SHM lexicon [2].

Arbitrariness in Labeling—Non-equivalence in Categories: One critical challenge highlighted in this study was the arbitrariness in defining diagnostic categories for data clusters. For instance, similar structural conditions were sometimes classified into distinct clusters due to subtle variations in preprocessing or feature selection. This highlights the non-equivalent diagnostic potential of different category definitions, reinforcing the need for a more standardized approach to labeling within SHM frameworks. Without addressing this arbitrariness, seemingly minor discrepancies could lead to significant interpretive divergences, impacting diagnostic reliability [7].

### 6.4. UAV Wing Monitoring

Arbitrariness—Variable Selection: The focus is on piezoelectric sensors, excluding temperature data, introduced arbitrariness, limiting interpretive possibilities [5,10].

### 6.5. Thermal Hot Spot Detection

Pragmatics—Contextual Bias in Sensor Use: Decisions to rely on piezoelectric sensors under specific conditions highlighted the role of pragmatics in shaping data interpretation [13,14].

The analyses of these projects reveal the critical influence of human decision making in shaping SHM methodologies. The works of Mujica and Ruiz emphasize the importance of creating standardized frameworks to minimize biases, ensuring a consistent ”language of data” across the field.

## 7. The Future of SHM: Toward a Universal Lexicon and Standardized Language of Data

Structural Health Monitoring (SHM) holds the potential for significant advancements through the development of standardized terminologies and rules for data collection, processing, and interpretation. Much like the evolution of grammatical rules in natural language, SHM depends on the organization and interpretation of data. A standardized lexicon and framework for SHM would improve accuracy, reliability, and cross-disciplinary collaboration across industries.

Farrar and Worden emphasize the importance of consistent definitions in SHM to ensure effective knowledge transfer and collaboration [15,16]. Similarly, Staszewski highlights the challenges posed by ambiguity in SHM terminology, underscoring the need for a structured, universal framework [17]. By creating shared rules and a unified lexicon, SHM can evolve into a system as reliable and universal as natural language.

### 7.1. Creating a Common Syntax: Regulating Data Collection

In both natural language and SHM, *syntax* plays a fundamental role in structuring information. Decisions about sensor placement, variable selection, and data collection form syntactic rules govern how data are processed and interpreted. Standardizing these decisions would create a unified system for data collection in SHM, ensuring consistency and reducing errors. Examples of such standardization include the following:Sensor placement: A unified framework for sensor placement across various structures would ensure that data from different projects can be meaningfully compared [15].Variable selection: Establishing standard criteria for selecting variables like stress, strain, and temperature will reduce biases and ensure that only relevant data are collected and analyzed [17].Labeling of data clusters: Developing consistent methods for labeling data clusters—based on categories with diagnostic significance—minimizes ambiguity and improves the interpretability of SHM results. This ensures the non-equivalent diagnostic potential for similar structural conditions is preserved [10].

### 7.2. Semantics of Data Interpretation: Standardizing Meaning

Semantics in SHM is analogous to deriving meaning from language. Standardized processes, such as Principal Component Analysis (PCA) and machine learning models, can ensure that data interpretation is consistent and reliable. Examples include the following:Universal models for dimensionality reduction: Shared models like PCA can distill key features from data, improving the consistency of interpretation [16].Benchmarking machine learning algorithms: Algorithms like K-Nearest Neighbors (KNN) should be evaluated against standard datasets to minimize biases and enhance reliability [15].

### 7.3. Pragmatics and Contextual Influence: Controlling for Environmental Factors

In linguistics, pragmatics considers how context influences meaning. Similarly, SHM requires models that account for environmental conditions (e.g., temperature and load) to improve diagnostics. Standardizing how contextual factors are measured and integrated into models is crucial [17].

### 7.4. Proposed Universal Lexicon for SHM

Developing a universal lexicon is essential to unify the terminology used in SHM. Key terms and definitions include the following:Damage identification: Detecting and characterizing changes in a structure that may affect its performance [15].Statistical pattern recognition: Utilizing statistical methods to identify patterns in data for damage detection [16].Structural degradation monitoring: Observing and assessing the deterioration of structural components over time [17].Machine learning in SHM: Algorithms that enable systems to learn from data and improve damage detection accuracy [16].Sensor technologies: Devices and methods used to collect data on structural integrity [15].Data preprocessing: Techniques for preparing raw data for analysis, including cleaning and normalization [17].Damage prognosis: Predicting the future condition and remaining useful life of a structure [15].Operational and environmental variability: Factors that influence structural performance and monitoring data [16].Feature extraction: Identifying relevant information from data that indicates structural health [17].Anomaly detection: Identifying deviations from normal behavior that may indicate damage [15].Category labeling: Assigning diagnostic categories to structural states or conditions based on the observed data. These labels often reflect subjective thresholds or domain-specific interpretations, leading to a non-equivalent diagnostic potential. Establishing criteria for consistent and context-aware labeling is critical for ensuring reliable diagnostics across similar structural conditions.

While these terms represent a shared foundation, variations in terminology persist across different studies and applications in SHM. For instance, “damage identification” might also be referred to as “structural anomaly detection”, depending on the context or focus of the research. Such inconsistencies can present challenges for communication and data interpretation within the SHM community.

Recognizing these variations, this section advocates for harmonizing terminology to facilitate clearer communication and collaboration. The proposed lexicon is not meant to be static but rather a starting point for a dynamic and inclusive dialogue within the SHM field. Researchers and practitioners are encouraged to provide feedback and contribute to refining and expanding this lexicon to ensure its comprehensiveness and applicability across various contexts.

By promoting a universal lexicon, this work aims to reduce ambiguities, enhance interdisciplinary collaboration, and foster a unified framework for SHM. This harmonization effort is essential for advancing both theoretical and practical aspects of the field.

### 7.5. Benchmarking and Standardization

Benchmarking underlines the need for uniform standards in SHM, allowing researchers to compare results and decide on shared criteria. For example, proposing standardized datasets and testing protocols in SHM can unify interpretations across studies, similar to the impact of grammar rules in language.

### 7.6. The Emergence of a Data Lexicon: Defining Neologisms in SHM

As SHM evolves, new terms and concepts will emerge to describe novel phenomena, much like neologisms in language. Some examples are as follows:A shared SHM lexicon: A glossary of standardized terms for describing defects, measurements, and algorithms will unify the SHM community [15].New categories for data clusters: As advanced sensors are developed, defining consistent categories for clustering similar structural states will help reduce the arbitrariness in diagnostic labeling. These categories will provide a more reliable framework for interpreting data across different conditions [16].

By addressing the issue of category arbitrariness explicitly, the proposed universal lexicon would ensure that the SHM community has a shared understanding of key diagnostic concepts. This clarity is essential not only for improving communication across disciplines but also for minimizing inconsistencies in data interpretation that arise from subjective labeling practices.

## 8. Scalability and Automation of AI in SHM

The objectives proposed in this study provide significant potential for improving the scalability and automation of Artificial Intelligence (AI) in Structural Health Monitoring (SHM). By standardizing the “language of data” and developing a universal lexicon, the framework addresses critical limitations in data consistency and interpretation. These advancements are the foundation for AI systems to operate more efficiently, generalize across multiple data sets, and reduce reliance on manual interventions in SHM processes.

### 8.1. Current Practices in Data Analysis by AI in SHM

At present, AI-based SHM involves several key steps in data analysis as follows:Data collection: Sensors installed on structures collect raw data, including vibration signals, strain measurements, and environmental variables. These datasets often lack standardization, resulting in inconsistent formats and representations.Preprocessing: Domain-specific preprocessing pipelines clean, normalize, and organize the data. However, these steps introduce variability, as preprocessing approaches differ significantly among practitioners.Feature extraction: Relevant features, such as principal components or modal parameters, are extracted to enable machine learning (ML) models to detect anomalies or classify structural states.Modeling: AI models, including supervised and unsupervised learning algorithms, are trained to identify patterns in the data and predict damage or degradation.Interpretation and decision making: The results produced by AI models are interpreted by experts, who make decisions based on the insights provided. This step remains semi-automated, often requiring substantial human input due to ambiguities in data interpretation.

### 8.2. Accelerating AI in SHM Through Standardization

By achieving a universal lexicon and standardizing the “language of data”, this study addresses key challenges in AI-driven SHM and enhances scalability and automation.

#### 8.2.1. Improving Scalability

Interoperability across systems: A universal lexicon ensures that data collected from different SHM systems can be seamlessly integrated, allowing AI models trained on one dataset to generalize effectively to new datasets without extensive retraining.Reusable preprocessing pipelines: Standardized preprocessing methods eliminate variability, enabling automated and consistent data preparation for various applications.Unified training datasets: Benchmark datasets aligned with the universal lexicon provide high-quality inputs for training AI models, improving accuracy and robustness across deployments.Consistency in category labeling: Standardized approaches to labeling data clusters address the arbitrariness inherent in category assignments. These assignments often vary based on subjective thresholds or context-specific criteria, introducing inconsistencies that reduce diagnostic reliability. By defining universal criteria for labeling, this effort ensures more consistent interpretations across applications.

#### 8.2.2. Enhancing Automation

Automated feature selection: Standardized data representations allow AI systems to automatically identify relevant variables, minimizing reliance on manual feature engineering.Real-time monitoring: Consistent data inputs enable real-time AI analysis, allowing SHM systems to detect anomalies and trigger alerts with minimal delay.Context-aware AI: Including environmental and operational variables in a standardized format allows AI systems to adapt to varying conditions, enhancing reliability.Reduction of diagnostic ambiguity: Addressing the arbitrariness in cluster labeling ensures that automated systems categorize structural states with greater precision, reducing potential errors stemming from subjective or non-standard labeling practices.

### 8.3. Generalization and Transfer Learning

Standardization facilitates the application of transfer learning techniques where pre-trained AI models are fine-tuned for new SHM applications. This reduces computational costs and accelerates the deployment of AI systems in diverse contexts. By eliminating ambiguities in data interpretation, including the arbitrariness of category labeling, AI systems achieve improved generalization across different structures, such as bridges, aircraft, and industrial equipment.

### 8.4. Strengthening Human–AI Collaboration

Transparent models: A universal lexicon ensures that AI outputs are interpretable, enabling experts to validate model decisions with greater confidence.Decision support systems: Automated analyses integrated into decision-making dashboards reduce cognitive load for engineers while providing clear, actionable insights.

### 8.5. Integrating Data Science, AI, and ML for SHM Scalability

This standardization effort strengthens the interconnection between data science, artificial intelligence, and machine learning, each of which plays a distinct role in SHM:Data science: Automated preprocessing pipelines and unified datasets streamline data preparation and exploratory analysis.AI: Scalable, context-aware AI models generalize across diverse applications, reducing development time and improving efficiency.ML: Enhanced feature extraction and transfer learning methods allow ML algorithms to perform more effectively on standardized inputs, improving prediction and classification accuracy.

Achieving the goals in this study brings a big change to SHM by turning time-consuming, custom-made solutions into automated and scalable systems. These improvements help make infrastructure monitoring clearer, more reliable, and easier to use. By solving problems with inconsistent and unclear data, this research opens the door to faster and more effective AI-based SHM methods.

## 9. Conclusions

Structural Health Monitoring (SHM) is a critical field dedicated to ensuring the safety, functionality, and longevity of structures. However, this paper has highlighted the inherent subjectivity present in SHM practices, framing biases and inconsistencies as elements of the “language of data”. This linguistic analogy reveals that data, much like human language, is not purely objective. Decisions regarding what to measure, how to measure it, and how to interpret results are inherently influenced by human judgment. Recognizing this subjectivity is essential for advancing SHM methodologies.

One particularly significant source of subjectivity in SHM is the arbitrariness involved in labeling data clusters. Categories assigned to clusters often vary based on context, diagnostic objectives, or thresholds defined by researchers. These labels are not neutral but carry implicit biases that influence diagnostic potential and outcomes. Recognizing the critical role of categories as sources of diagnostic divergence, this study emphasizes the need to address such arbitrariness within a standardized lexicon. Establishing universally agreed-upon criteria for labeling clusters would significantly enhance consistency and diagnostic reliability.

The traditional perception of data as a neutral and unbiased reflection of reality, deeply rooted in scientific epistemology, has been challenged by perspectives from fields such as constructivism, cybernetics, quantum physics, and language ontology. These disciplines emphasize the subjective and interpretive nature of language—including the “language of data”. This study applies these insights to SHM, arguing that the methods and tools used to collect and interpret data actively shape the conclusions drawn. These human-influenced processes, including cluster labeling, introduce ambiguity and variability that must be managed to enhance SHM reliability.

A key contribution of this paper is the proposal of a universal SHM lexicon—a standardized “grammar” for interpreting data. By creating a shared language that defines key concepts such as damage identification, statistical pattern recognition, anomaly detection, and category labeling for clusters, the SHM community can reduce ambiguities, improve interdisciplinary communication, and facilitate collaboration. The proposed lexicon, inspired by foundational works such as those by Farrar, Worden, and Staszewski, aims to unify the diverse practices and terminologies currently used in SHM.

The potential benefits of adopting a standardized SHM lexicon are substantial and outlined as follows:Improved diagnostic accuracy: By minimizing subjective biases in data collection, cluster labeling, and interpretation, SHM systems can provide more accurate and reliable diagnoses.Enhanced collaboration: A common language facilitates interdisciplinary and international cooperation, enabling researchers and practitioners to share knowledge more effectively.Scalability and automation: Standardized terminology and methodologies are crucial for integrating SHM into automated systems, including AI and machine learning models.Educational impact: A universal lexicon provides a foundation for training future SHM professionals, ensuring consistency and clarity in the field.

Furthermore, the study underscores the importance of acknowledging and managing the subjective aspects of SHM. While complete objectivity may be unattainable, strategies such as benchmarking, data fusion, and context-aware modeling can mitigate the impact of biases. Addressing arbitrariness in cluster labeling is a crucial step in this process, ensuring diagnostic categories better reflect structural realities and reduce variability in outcomes. Recognizing the limits of human interpretation allows researchers to design systems that are more transparent, robust, and inclusive of environmental and operational variability.

Looking ahead, the next steps in advancing SHM as a standardized discipline include the following:Developing and validating the proposed lexicon through interdisciplinary collaboration and case studies.Establishing international working groups to refine and formalize SHM standards.Leveraging AI and machine learning to operationalize the lexicon in real-time monitoring systems, with particular emphasis on reducing the arbitrariness in labeling data clusters.Promoting the lexicon through education and training initiatives to ensure widespread adoption.

In conclusion, this paper provides a conceptual framework for interpreting data in SHM as a form of language. By standardizing this “language of data”, the SHM community can address the challenges of ambiguity, bias, and inconsistency, paving the way for more reliable, efficient, and collaborative structural diagnostics. Recognizing and managing the inherent subjectivity of data, including the critical role of arbitrary categories in cluster labeling, is not a limitation but an opportunity to build a more transparent and effective SHM ecosystem. The integration of a universal lexicon into SHM practices has the potential to transform the field, advancing both theoretical understanding and practical applications.

## Figures and Tables

**Table 1 sensors-25-03054-t001:** Challenges in data science and the need for a structured data language.

Aspect	Current Problem	Proposed Solution (Data Language)
**Bias in Machine Learning**	Arbitrary data categorization introduces distortions in models and reduces their generalization ability.	Define a clear semantic framework for data, eliminating subjective interpretations.
**Interoperability**	Lack of standards in data organization prevents efficient model transfer across disciplines.	Structure data with shared syntactic rules to improve compatibility.
**Automation in** **Data Science**	AI cannot operate effectively with ambiguous or inconsistently structured information.	Create a unified syntax, semantics, and pragmatics framework that enables algorithms to interpret data as a language.
**Replicability in** **Studies**	Subjectivity in variable selection and labeling makes comparisons between research studies difficult.	Apply a standardized data language to ensure consistent interpretation across studies.

## Data Availability

This work builds on previously published studies. No new data were generated during the study. We have cited our own work (Mujica and Ruiz) on SHM, providing a critical analysis of how each decision, including class labeling and other aspects, is influenced by the individual experience and perception of the experts guiding the research.

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
