# Peer review of "Data Interpretation in Structural Health Monitoring: Toward a Universal Language"

_sensors, 2025, doi:10.3390/s25103054_

Round 1
Reviewer 1 Report
Comments and Suggestions for Authors
By reading the article a reader immediately understand the importance of a common language between researchers and SHM professionals to avoid misunderstandings, inaccuracies, dependence on subjective evaluations, etc. The article, therefore, lists all the requirements that, in opinion of the Authors, this language should have, also highlighting what type of language should be used in the various phases of the SHM (data collection, diagnostic phase, real time monitoring phase. Finally, the article closes with all the steps necessary to obtain this universal language for SHM.
It all seems, therefore, to be a "framework" description of a work yet to be carried out. The reader, in my opinion, therefore cannot understand what is actually intended to be done in concrete terms. I therefore suggest inserting a paragraph with an example taken from experimental reality, in which the authors show this language, perhaps not yet perfect, but at least sketchy. Similarly to an experimental article, after having presented all the theoretical apparatus behind a type of experiment, but then one is actually carried out and it is shown how it is correct to use the theoretical apparatus.
Author Response
Reviewer 1 – Concrete Example and Applicability
Comment:
The manuscript feels like a framework proposal for a future work. It lacks a concrete example that demonstrates the applicability of the theoretical framework.
Response:
We fully agree with this observation. In response, we have several real-world cases studies, which illustrate how subjective data categorization can influence structural diagnostics. Rather than selecting an external case, we deliberately chose to reflect on our own prior work. This decision was made because our intention is not to critique the efforts of others but to spark an open discussion by analyzing what is often overlooked in our own practice. The cases demonstrate how expert-dependent choices in class labeling, variable selection, and interpretation introduce variability, supporting our call for a structured, language-based approach to SHM data.
Reviewer 2 Report
Comments and Suggestions for Authors
(1) The author appears to have engaged in some popular science work. This article seems familiar to anyone who has undergone health monitoring. However, I fail to see the significance of this article. It discusses various data processing methods, but it is unclear what the author aims to achieve with this article. Is it a review? What is its innovation? I suggest dedicating a section to explicitly outlining the purpose and novelty of the article. As it stands, I find it difficult to assess the value of the manuscript.
(2) You have included a great deal of popular science information and seem to be attempting to compile it into a coherent narrative. While this is an ambitious goal, I am unfortunately disappointed with the content. I feel that I have not derived any useful insights from your paper. I strongly recommend revising these sections to clearly demonstrate their value.
(3) You have presented numerous data processing methods; however, you have failed to cite relevant references to support your claims.
(4) Your introduction is weak. It does not adequately explain the current state of research nor highlight the significance of the study.
(5) The abstract does not appear to reflect the significance of your research.
(6) If your intention is to write a review, I strongly recommend changing the title to better reflect this aim.
Author Response
Response to Reviewer 2
Comment 1: The article lacks a clear statement of purpose and novelty.
Response: We have added a dedicated section titled “Contributions and Novelty of this Work” that outlines the conceptual and practical innovations of our proposal. It clearly distinguishes this work from previous literature by introducing a linguistically inspired approach to data interpretation and standardization in SHM.
Comment 2: There is too much “popular science” language and little scientific value derived from the content.
Response: We have revised the manuscript throughout to ensure that all claims are supported by appropriate scientific literature and references. The tone has been adjusted to match the expectations of a technical and research-oriented readership. Furthermore, the theoretical content has been expanded with citations from the SHM community and recent contributions in the field of epistemology, constructivism, and data interpretation.
Comment 3: The introduction and abstract do not reflect the significance of the work.
Response: Both the abstract and introduction have been rewritten to emphasize the motivation, the current gap in the literature, and the value of addressing data ambiguity and arbitrariness in SHM diagnostics.
Comment 4: If the manuscript is intended as a review, the title should reflect that.
Response: While the paper discusses existing issues in SHM, it is not structured as a review article. Its intent is to propose a new framework, which is now made explicit in both the title and abstract. We hope the current version more accurately reflects the nature of the manuscript.
Round 2
Reviewer 1 Report
Comments and Suggestions for Authors
no notes
Reviewer 2 Report
Comments and Suggestions for Authors
Accept